METHODS AND RESOURCES

# MosAIC: An annotated collection of mosquito-associated bacteria with high-quality genome assemblies

Aidan Foo[1☉], Laura E. Brettell[1,2☉], Holly L. Nichols[3☉], 2022 UW-Madison Capstone in Microbiology Students[¶], Miguel Medina Muñoz[3], Jessica A. Lysne[3], Vishaal Dhokiya[1], Ananya F. Hoque[1], Doug E. Brackney[4,5], Eric P. Caragata[6], Michael L. Hutchinson[7,8], Marcelo Jacobs-Lorena[9], David J. Lampe[10], Edwige Martin[11], Claire Valiente Moro[11], Michael Povelones[12], Sarah M. Short[13], Blaire Steven[14], Jiannong Xu[15], Timothy D. Paustian[3], Michelle R. Rondon[3], Grant L. Hughes[1,16], Kerri L. Coon[3‡]*, Eva Heinz[1,17,18]*

1 Department of Vector Biology, Liverpool School of Tropical Medicine, Liverpool, United Kingdom, 2 School of Science, Engineering and Environment, University of Salford, Manchester, United Kingdom, 3 Department of Bacteriology, University of Wisconsin-Madison, Madison, Wisconsin, United States of America, 4 Department of Entomology, Connecticut Agricultural Experiment Station, New Haven, Connecticut, United States of America, 5 Center for Vector Biology and Zoonotic Diseases, Connecticut Agricultural Experiment Station, New Haven, Connecticut, United States of America, 6 Florida Medical Entomology Laboratory, Department of Entomology and Nematology, Institute of Food and Agricultural Sciences, University of Florida, Vero Beach, Florida, United States of America, 7 Division of Vector Management, Pennsylvania Department of Environmental Protection, Harrisburg, Pennsylvania, United States of America, 8 Division of Plant Health, Pennsylvania Department of Agriculture, Harrisburg, Pennsylvania, United States of America, 9 Department of Molecular Microbiology and Immunology, Malaria Research Institute, Johns Hopkins Bloomberg School of Public Health, Baltimore, Maryland, United States of America, 10 Department of Biological Sciences, Duquesne University, Pittsburgh, Pennsylvania, United States of America, 11 Universite Claude Bernard Lyon 1, Laboratoire d'Ecologie Microbienne, UMR CNRS 5557, UMR INRAE 1418, VetAgro Sup, 69622 Villeurbanne, France, 12 Department of Pathobiology, School of Veterinary Medicine, University of Pennsylvania, Philadelphia, Pennsylvania, United States of America, 13 Department of Entomology, The Ohio State University, Columbus, Ohio, United States of America, 14 Department of Environmental Science and Forestry, Connecticut Agricultural Experiment Station, New Haven, Connecticut, United States of America, 15 Department of Biology, New Mexico State University, Las Cruces, New Mexico, United States of America, 16 Department of Tropical Disease Biology, Centre for Neglected Tropical Disease, Liverpool School of Tropical Medicine, Liverpool, United Kingdom, 17 Department of Clinical Sciences, Liverpool School of Tropical Medicine, Liverpool, United Kingdom, 18 Strathclyde Institute of Pharmacy and Biomedical Sciences, University of Strathclyde, Glasgow, United Kingdom

☉ These authors contributed equally to this work.
‡ To whom correspondence regarding access to isolates as well as future expansions to the collection should be addressed: kerri.coon@wisc.edu
¶ Membership of 2022 UW-Madison Capstone in Microbiology Students is provided in S1 File.
* kerri.coon@wisc.edu (KLC); eva.heinz@strath.ac.uk (EH)

**Data Availability Statement:** Physical isolates for all 392 bacterial strains in the collection have been deposited for long-term preservation and public

## Abstract

Mosquitoes transmit medically important human pathogens, including viruses like dengue virus and parasites such as *Plasmodium* spp., the causative agent of malaria. Mosquito microbiomes are critically important for the ability of mosquitoes to transmit disease-causing agents. However, while large collections of bacterial isolates and genomic data exist for vertebrate microbiomes, the vast majority of work in mosquitoes to date is based on 16S rRNA gene amplicon data that provides limited taxonomic resolution and no functional information. To address this gap and facilitate future studies using experimental microbiome

access in the American Type Culture Collection (ATCC; Gaithersburg, MD, USA). Isolates are also available upon request (see https://kcoonlab.bact. wisc.edu/mosaic/). Raw Illumina reads and assemblies are available in the NCBI Sequence Read Archive (https://www.ncbi.nlm.nih.gov/sra) under BioProject ID PRJNA1023190 (see accessions in S1 Table). Scripts used for analysis and figure generation, along with relevant underlying data (e.g., newick tree files), are available in MosAIC's GitHub Repository (https:// github.com/MosAIC-Collection/MosAIC_V1) under Zenodo DOI: https://zenodo.org/doi/10.5281/ zenodo.13934797. All other relevant data supporting the findings of this study, including isolate-specific metadata, are available within the article and its supplementary information. A README file outlining how repository code/data can be used to reproduce all figures in the manuscript is provided in S2 File.

**Funding:** This work was supported by collaborative awards from the National Science Foundation (NSF, https://www.nsf.gov/, IOS-2019368 to KLC) and the Biotechnology and Biological Sciences Research Council (BBSRC, https://www.ukri.org/ councils/bbsrc/, BB/V011278/1 and BB/V011278/2 to EH) and the National Institutes of Health (NIH, https://www.nih.gov/) (R21AI138074) (to GLH). This work was further supported by BBSRC (BB/ T001240/1, BB/X018024/1, and BB/W018446/1 to GLH), the UK Research and Innovation (UKRI, https://www.ukri.org/) (20197 and 85336 to GLH), the Engineering and Physical Sciences Research Council (https://www.ukri.org/councils/epsrc/) (V043811/1 to GLH), a Royal Society Wolfson Fellowship (https://royalsociety.org/)(RSWF\R1 \180013 to GLH), the National Institute for Health and Care Research (https://www.nihr.ac.uk/) (NIHR2000907 to GLH), and the Bill and Melinda Gates Foundation (https://www.gatesfoundation. org/) (INV-048598 to GLH). KLC was further supported by the U.S. Department of Agriculture (https://www.usda.gov/) (2018-67012-29991). EH was further supported by the Wellcome Trust (https://wellcome.org/) (217303/Z/19/Z). LEB was supported by the Liverpool School of Tropical Medicine Director's Catalyst Fund (https://www. lstmed.ac.uk/). AF and VD were supported by the UKRI Medical Research Council (https://www.ukri. org/councils/mrc/)(MR/N013514/1). HLN was supported by the NSF (DGE-2137424) and JAL was supported by the NIH (R25GM144251). The funders had no role in study design, data collection and analysis, decision to publish, or preparation of the manuscript.

manipulations, we generated a bacterial Mosquito-Associated Isolate Collection (MosAIC) consisting of 392 bacterial isolates with extensive metadata and high-quality draft genome assemblies that are publicly available, both isolates and sequence data, for use by the scientific community. MosAIC encompasses 142 species spanning 29 bacterial families, with members of the *Enterobacteriaceae* comprising 40% of the collection. Phylogenomic analysis of 3 genera, *Enterobacter*, *Serratia*, and *Elizabethkingia*, reveal lineages of mosquito-associated bacteria isolated from different mosquito species in multiple laboratories. Investigation into species' pangenomes further reveals clusters of genes specific to these lineages, which are of interest for future work to test for functions connected to mosquito host association. Altogether, we describe the generation of a physical collection of mosquito-associated bacterial isolates, their genomic data, and analyses of selected groups in context of genome data from closely related isolates, providing a unique, highly valuable resource for research on bacterial colonisation and adaptation within mosquito hosts. Future efforts will expand the collection to include broader geographic and host species representation, especially from individuals collected from field populations, as well as other mosquito-associated microbes, including fungi, archaea, and protozoa.

## Introduction

Mosquitoes (Diptera: Culicidae) are the major vectors of some of the world's most important pathogens. The mosquito microbiome can influence all aspects of host biology [1], including the mosquito's ability to transmit human pathogens [2–9]. Given these functions and the increasing resistance of mosquitoes against insecticides [10,11], microbiome manipulation is a promising and increasingly relevant alternative avenue for future applications in vector control. It is thus of urgent relevance to gain a better understanding of the mosquito microbiome's functional composition and its interactions with the host and invading pathogens.

The composition of the mosquito microbiome is dynamic and affected by host species [12,13] and geography [14,15], and varies across individuals [16] as well as across an individual's life stages [17–19]. Nevertheless, 16S rRNA gene amplicon sequencing studies have identified a number of commonly present bacterial genera, including *Enterobacter*, *Serratia*, *Asaia*, *Pantoea*, *Elizabethkingia*, and *Cedecea* [16,19–21]. Results from amplicon sequencing are however limited in taxonomic resolution and provide no functional information [22], rendering it challenging to draw conclusions beyond presence-absence. Experimental manipulations have demonstrated the capacity of specific bacterial isolates to influence mosquito life history in diverse ways. For example, *Serratia marcescens* has been shown to suppress adult feeding behaviour [23,24], while *Asaia* has been shown to accelerate larval development [25,26] and activate mosquito immune genes [27]. *Serratia marcescens* and *Enterobacter cloacae* have further been shown to positively influence both adult longevity and egg hatch rates in *Aedes aegypti* [28]. Another study showed *S. marcescens* to have larvicidal properties [29], highlighting the need for strain-level resolution to fully understand mosquito–microbiome interactions.

Understanding why certain bacteria are successful colonisers and how the host and microbial community interact will facilitate development of microbe-based approaches for vector control. This includes strategies like paratransgenesis, where genetically modified bacteria are introduced into the host to express effector molecules to block pathogen transmission or

**Competing interests:** The authors have declared that no competing interests exist.

**Abbreviations:** ANI, average nucleotide identity; GTDB, Genome Taxonomy Database; LSTM, Liverpool School of Tropical Medicine; MosAIC, Mosquito-Associated Isolate Collection; PBS, phosphate-buffered saline; SNP, single-nucleotide polymorphism; VFDB, virulence factor database.

interfere with mosquito longevity or reproduction [30–33]. In addition to their medical relevance, mosquitoes are also attractive experimental organisms, as they are amenable to microbiome manipulation through the use of germ-free and gnotobiotic techniques. One such approach involves hatching surface sterilised eggs to produce axenic (microbe-free) larvae that can then be seeded with a microbial community through inoculation of larval water [1,34–36]. Further, complete mosquito microbiomes can be successfully extracted, cryopreserved [37], and transplanted between different mosquito species [38]. Using these approaches, members of our team recently determined the ability of *Cedecea neteri* to form biofilms, showing this to be a key factor that contributes to colonisation of *Aedes aegypti* mosquitoes [39]. The transcriptome of transplant recipients was also demonstrated to respond similarly when receiving a microbiome from a laboratory-reared donor regardless of mosquito species, but showed more differentially expressed genes when the donor was field-caught [40].

The taxonomic composition of the mosquito microbiome and its effects on mosquito life history are well characterised in the literature [41–50]. However, our understanding of factors that drive bacterial colonisation in the mosquito remain poorly understood. Virulence factors are genes that contribute to bacterial establishment and persistence within a host, including insects [51], and their identification is facilitated by use of whole genome analyses. For example, comparative genomics of *Serratia* isolated from different mosquito species shows differing virulence factor profiles, suggesting differences in their ability to infect the mosquito host [52]. Indeed, within other fields, insights into symbiont colonisation and host interactions are facilitated by large-scale dedicated genomic collections of bacterial isolates [53–57]—a resource currently absent in the mosquito microbiome field.

With the aim of facilitating future mosquito microbiome studies by the scientific community at large, we report the generation of a collection of mosquito-associated bacterial isolates with accompanying sample metadata and high-quality assembled genomes. Pangenome analyses further indicate the presence of mosquito-associated lineages within 3 focal genera, *Enterobacter*, *Serratia*, and *Elizabethkingia*. Our results identify highly conserved lineage-specific core genes that represent promising candidates for future studies to explore potential bacterial interactions with mosquito hosts.

## Results

### 392 draft genomes assembled from the mosquito microbiome and associated environments

MosAIC represents a collection of 392 bacterial isolates and high-quality draft genomic assemblies, including 169 isolates derived from laboratory reared mosquitoes, 126 isolates derived from field caught mosquitoes, 83 isolates derived from mosquito larval habitats, and 14 isolates derived from non-mosquito Diptera and their associated environments sampled from both the laboratory and field (Figs 1 and 2A and S1 and S2 Table). *Aedes aegypti* is the most commonly represented mosquito host species, for which 112 isolates were cultured from eggs, larvae, or adults, followed by *Aedes albopictus* (*n* = 71), *Culex pipiens* (*n* = 23), *Anopheles quadrimaculatus* (*n* = 19), *Anopheles gambiae* (*n* = 15), *Aedes triseriatus* (*n* = 12), *Toxorhynchites amboinensis* (*n* = 10), *Culex nigripalpus* (*n* = 9), *Aedes atropalpus* (*n* = 5), *Aedes taeniorhynchus* (*n* = 5), *Deinocerites cancer* (*n* = 4), *Anopheles crucians* (*n* = 3), *Anopheles* sp. (*n* = 2), *Aedes trivittatus* (*n* = 1), *Anopheles punctipennis* (*n* = 1), and *Culex erraticus* (*n* = 1) (Fig 1 and S2 Table). As it is only female mosquitoes that transmit pathogens, MosAIC is heavily skewed towards female-derived isolates. Of the 245 total isolates recovered from adult mosquitoes, 219 were derived from females. Across all adults, the majority were obtained from non-blood-fed (*n* = 168) as compared to blood-fed females (*n* = 51) (Fig 1 and S2 Table). Adult-derived

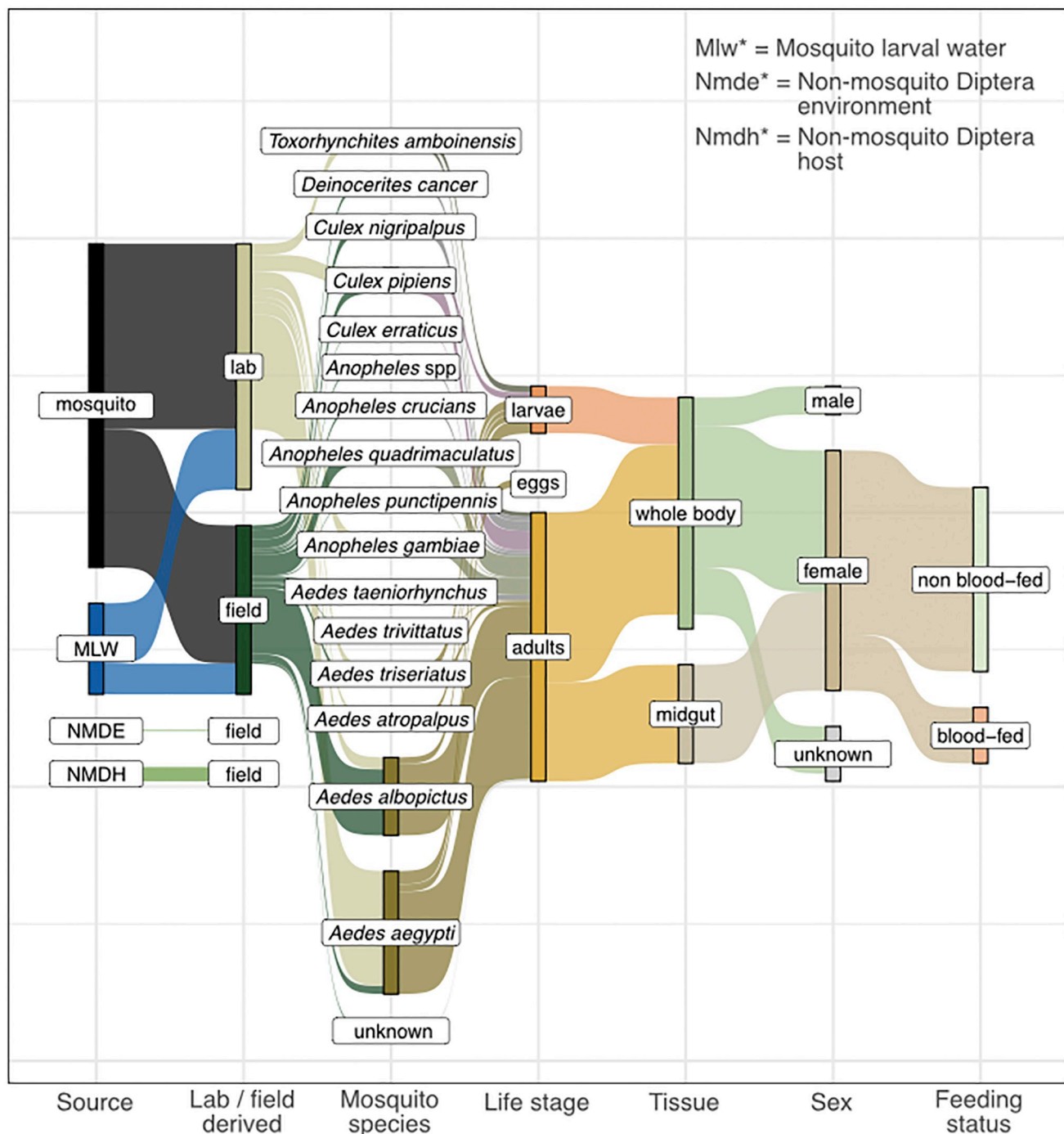

**Fig 1. Origin of bacterial isolates in MosAIC.** Metadata category names and definitions follow those presented in S1 Table. "Unknown" denotes isolates for which a given metadata category is valid but missing. For example, a subset of mosquito samples could not be assigned a species but are derived from adult-stage mosquitoes. Where a given metadata category is invalid, the connection between bars is dropped. For example, feeding status is not a valid category for egg samples. All code and data to recreate this figure can be found at https://github.com/MosAIC-Collection/MosAIC_V1 in the folder "04_Sankey_Diagram." MosAIC, Mosquito-Associated Isolate Collection.

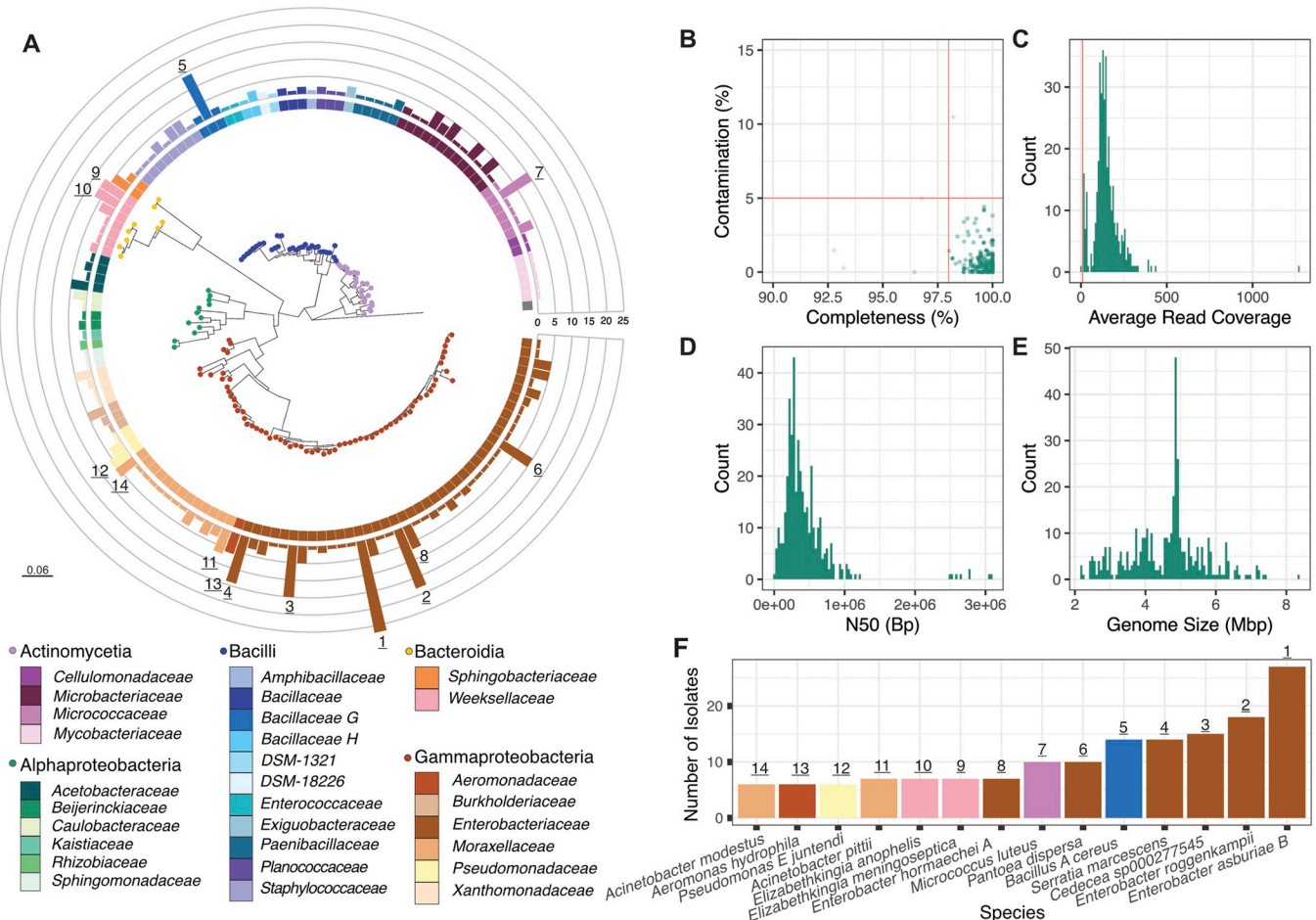

**Fig 2. Phylogeny of single species representatives from MosAIC, along with quality-assurance metrics for related genome assemblies.** (A) Maximum likelihood tree built using IQ-TREE2 and 16S rRNA gene sequences predicted with Baarnap. Each node is a species representative coloured according to class. Bars at each tip represent the number of isolates present in the species cluster, defined using a secondary clustering threshold of 95% ANI with dRep. Bars are colour coded according to family information obtained using the Genome Taxonomy Database and classifier GTDB-Tk. Numbers at the tip of bars delineate highly representative species clusters. Evolutionary scale is displayed on the bottom left of the figure panel. (B) Genome completeness and contamination metrics obtained using CheckM. Each point represents a draft genome assembly. Red lines indicate cutoffs for 98% completeness and 5% contamination. (C) Histogram showing average read coverage reported using QUAST. The vertical red line represents a 10× filter cutoff. (D, E) Histograms showing N50 values (the length of the shortest sequence within a group of sequences that represent 50% of the overall assembly) and genome size across the collection. Bars represent high-quality genomes within the collection (CheckM completeness >98%, contamination <5%, and >10X coverage). Bp = Base-pairs, Mbp = Mega base-pairs. (F) Number of isolates comprising the highly represented species (>5 isolates) within the collection. Each bar is coloured according to family and numbered according to their placement in the main phylogeny in panel (A). All code and data to recreate this figure can be found at https://github.com/MosAIC-Collection/MosAIC_V1. For Fig 2A, the code and data are in the folder "03_MosAIC_Phylogeny;" for Fig 2B–2E, they are in the folder "01_GenomeQC," and for Fig 2F, they are in the folder "02_GTDB_Drep_Summary".

isolates were also predominantly recovered from whole-body (*n* = 155) as compared to midgut samples (*n* = 90) (Fig 1 and S2 Table).

Altogether, we sequenced in total 82 GB of read data containing 4.2 megabase pairs (Mbp) on average per sample, ranging between 0.26 Mbp and 27.1 Mbp (S1 Fig). After quality-checking reads, assemblies were assessed based on single-copy core gene content and average genome coverage. The mean genome completeness score across the collection is 99.67% (98.01%–100%) and the mean contamination score is 0.59% (0%–4.40%) (Fig 2B). Genomes in the collection have an average coverage of >10×, range in size between 2.21 and 8.38 Mbp, and are characterised by N50 values ranging between 9.14 kilobase pairs (kbp) and 3.09 Mbp

(Fig 2C–2E). Genome sizes and the number of predicted genes of isolates in the collection are also linearly correlated (S2 Fig) [58].

## MosAIC highlights species representatives of known mosquito-associated bacteria

The Genome Taxonomy Database (GTDB) was used to predict the taxonomy of isolates (S3 Table). Sixty-two genomes shared >99% average nucleotide identity (ANI) with GTDB reference genomes, while 305 shared between 95% and 99% ANI and 25 shared <95% ANI, potentially representing novel species [59]. As such, 367 isolates were assigned to a species, with the remaining 25 genomes assigned to their respective genera. Of these 25 genomes, 6 isolates were assigned to *Leucobacter*, 6 isolates were assigned to *Microbacterium*, and the remaining 12 isolates were each assigned to individual genera (S3 Fig).

Summarising the collection into species clusters using dRep resulted in 142 single species representatives (Fig 2A) in 5 bacterial classes: Actinomycetes (4 families), Alphaproteobacteria (6 families), Bacilli (11 families), Bacteroidia (2 families), and Gammaproteobacteria (6 families) (Fig 2A). The collection is dominated by Gammaproteobacteria; in particular, the *Enterobacteriaceae* comprised 42 species representatives and 157 isolates. The *Microbacteriaceae* and *Weeksellaceae* are also well represented, with 31 and 21 isolates, respectively. As such, 6/10 most well-represented species in the collection are *Enterobacteriaceae*, with the remainder belonging to *Bacillaceae*, *Micrococcaceae*, *Moraxellaceae*, and *Weeksellaceae* (Fig 2F).

## MosAIC isolates are distributed across a range of different sample types

MosAIC is accompanied by comprehensive metadata associated with each isolate (S1 Table). As this initial collection was derived from existing samples and not under any specific sampling framework, it is not possible to draw conclusions regarding bacterial taxa that may be associated exclusively with particular sample types or conditions. However, we do note some interesting observations from the collection presented to date, including the observation that some taxa, including members of the *Acetobacteraceae*, *Bacillaceae_G*, and *Caulobacteraceae*, among others, were only isolated from non-blood-fed female and male adult mosquitoes, while members of other taxa such as *DSM-1321* were only isolated from blood-fed females (S4A Fig). Taxa were well represented from both laboratory- and field-derived mosquitoes across adult, larval, larval water, and egg samples, but there were differences between the overall number of isolates between these 2 sources. For example, 78% (*n* = 21) of isolates assigned to the bacterial families *Bacillaceae*, *Bacillaceae_G*, or *Bacillaceae_H* were from field-derived samples, whereas 84% (*n* = 26) of isolates assigned to the *Microbacteriaceae* were from laboratory-derived samples (S4B Fig). In contrast, the most dominant family represented in the collection, *Enterobacteriaceae*, was split relatively equally with 48% (*n* = 75) of isolates assigned to this family originating from field-derived samples and 52% (*n* = 82) from laboratory-derived samples (S4B Fig). Other taxa, including members of the *Cellulomonadaceae*, were isolated from adult male but not adult female mosquitoes (S4C Fig), while members of the *Rhizobiaceae* and *Exiguobacteraceae* were only isolated from midgut samples (S4D Fig).

We also note interesting patterns among genera within the *Enterobacteriaceae*—the most abundant and frequently observed family across MosAIC. Of laboratory- and field-derived isolates, we note 91% (*n* = 31) of *Pantoea* isolates were from field-derived samples, whereas 80% of *Enterobacter* (*n* = 44) isolates were laboratory-derived (S5A Fig). *Atlantibacter*, *Cedecea*, *Chania*, and *Rouxiella* were exclusively laboratory-derived (S5A Fig), and 57% (*n* = 90) of *Enterobacteriaceae* were isolated from female adult mosquitoes, while only *Pantoea* and *Enterobacter* were isolated from both male and female adult mosquitoes (S5B Fig).

## Mosquito-associated bacteria contain known and predicted virulence factors

We screened the collection for known and predicted virulence factor genes using the virulence factor database (VFDB). Gene hits were categorised using the VFDB scheme and therefore divided into the VFDB categories adherence, antimicrobial activity/competitive advantage, biofilm, effector delivery system, exoenzyme, exotoxin, immune modulation, invasion, motility, nutritional/metabolic factor, regulation, stress survival, and other (Fig 3).

In total, we identified 11,774 virulence factor genes across the collection and 1,203 genes that appeared in only 1 sample (S4 Table). The mean number of virulence genes was highly variable among bacterial classes ranging from Gammaproteobacteria and Bacilli, which contained on average 45 and 52 virulence genes, respectively, to Actinomycetes and Alphaproteobacteria, which contained on average 3 and 1 virulence genes, respectively, and Bacteroidia, in

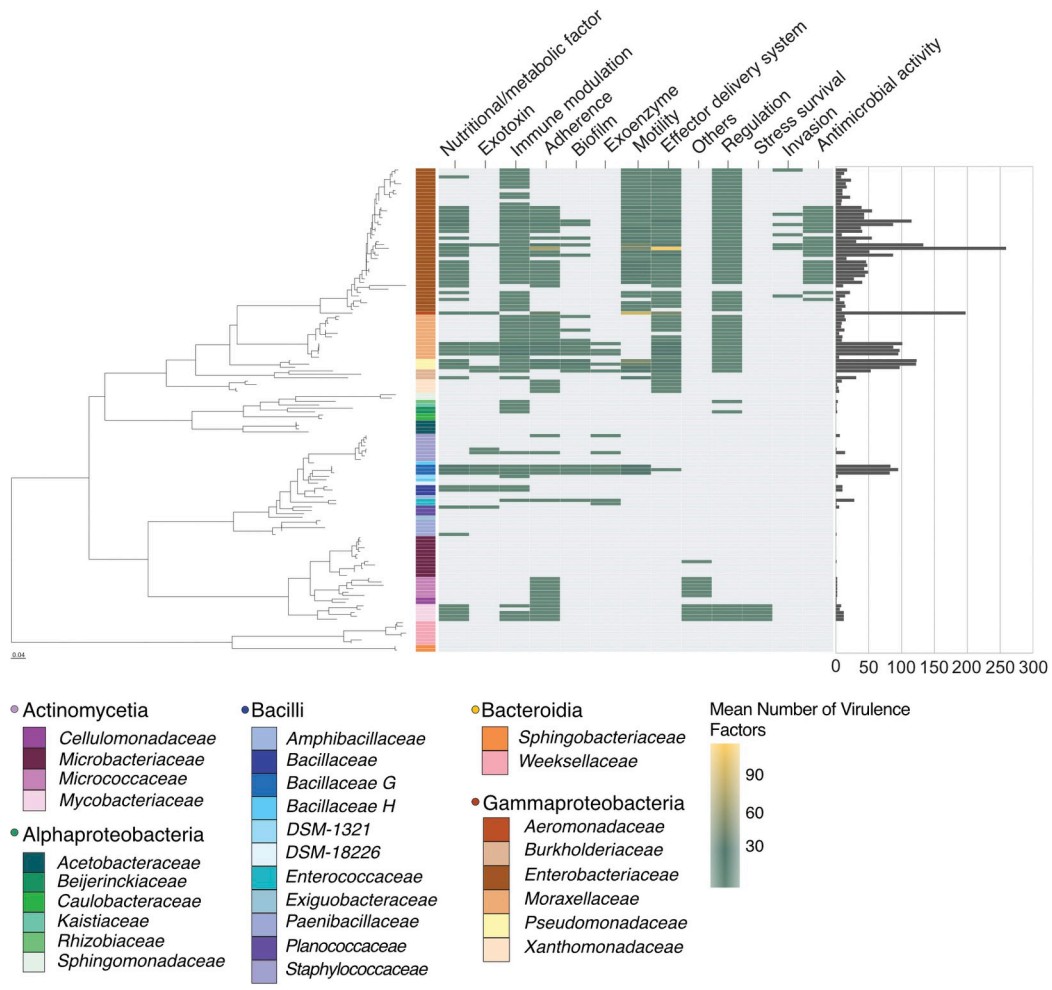

**Fig 3. Heatmap of the distribution of virulence factors across all MosAIC genomes.** Genes fall within one of 13 different categories (*top*). The guidance tree on the left is a maximum likelihood tree built using IQ-TREE2 and Baarnap-predicted 16S rRNA gene sequences from species clusters defined with dRep. Tiles denote the mean number of virulence factor genes identified within a given species cluster, following a gradient from blue (low) to yellow (high). Grey tiles denote species clusters for which zero predicted virulence factor genes were identified. Bacterial families are colour-coded in the figure legend. The bar chart on the right shows the total number of genes identified within each species cluster. All code and data to recreate this figure can be found at https://github.com/MosAIC-Collection/MosAIC_V1 in the folder "05_Virulence_Factor_Analysis." MosAIC, Mosquito-Associated Isolate Collection.

which we did not detect any virulence genes (Fig 3). The distribution of virulence factor categories also varied between bacterial classes. For example, stress survival genes were found only in the Actinomycetes. Similarly, genes involved in invasion and antimicrobial activity were only detected in the Gammaproteobacteria. Within bacterial families, there was also variation in virulence factor profiles; genes involved in stress survival were only detected in the *Mycobacteriaceae*, and genes involved in invasion and antimicrobial activity were common in the *Enterobacteriaceae* but were not detected in any other family (Fig 3). On the other hand, genes implicated in roles such as adherence and nutritional/metabolic factors were widespread and present in multiple families. It must be noted, however, that detection of these virulence factors is constrained by the inherent bias of the database composition in that there is limited representation of certain bacterial classes, including a complete absence of genes in the Bacteroidia (S6 Fig).

## Placement of mosquito-associated isolates into wider population structures

To determine the placement of MosAIC isolates in their wider population structures, we selected 3 taxonomic groups for further analysis (*Enterobacter*, *Serratia*, and *Elizabethkingia anophelis*), owing to their high representation within the collection, biological interest among the mosquito microbiome field [34], and good understanding of their genetic diversity and environmental/host-associated niches within a population (i.e., their population structure) [60–62]. In total, 55 *Enterobacter* isolates, 16 *Serratia* isolates, and 7 *El. anophelis* isolates were phylogenetically placed into their respective populations (S7–S9 Figs), which comprise previously sequenced isolates from a variety of environmental niches, including human bodily fluids, plants and soils, samples from mosquitoes, other insects and higher mammals, and water from a range of sources (S5 Table). The placement of these mosquito-associated genomes within their larger species clades helped refine GTDB-taxonomic classifications; *Serratia nevei* and *Serratia bockelmannii* are now within the *S. marcescens* clade (Fig 4D), and a single *Enterobacter hormaechei* isolate falls within the *Enterobacter xiagfangensis* clade (Fig 4A).

Population structures revealed the clustering of mosquito-associated isolates within each of these taxa (*Enterobacter*, *Serratia*, and *Elizabethkingia anophelis*) (Fig 4A, 4D, and 4G). For *Enterobacter*, the majority of mosquito-associated isolates grouped with *Enterobacter asburiae* and *Enterobacter rogennkampii*, with MosAIC isolates forming several distinct lineages comprised of 21 and 18 isolates, respectively (Fig 4B and 4C). For *Serratia*, most mosquito-associated isolates belonged to *S. marcescens* and *S. fonticola*, including monophyletic lineages of 7 and 3 isolates, respectively (Fig 4E and 4F). The largest mosquito-associated lineage of *En. asburiae* contained isolates from 2 different mosquito species (*Ae. aegypti*, *An. gambiae*) that were isolated by 2 different, geographically distinct, laboratories (research groups) (Fig 4B). In contrast, the *En. roggenkampii* mosquito-associated lineage consisted of isolates from laboratory-reared *Ae. aegypti* mosquitoes maintained by a single laboratory (Fig 4C). Within the major *S. marcescens* lineage containing mosquito-derived isolates, there are 7 isolates derived from 3 mosquito species (*Ae. aegypti*, *An. gambiae*, and *An. stephensi*,) and 4 different laboratories among a large number of isolates derived from human and clinical samples (Fig 4E). Of these 7 isolates, 4 are from MosAIC and 3 originate from previous studies [52]. A similar observation can be seen for *El. anophelis*, where MosAIC-derived isolates from 3 different mosquito species and 2 different laboratories form a monophyletic group (Fig 4G and 4H).

Not all mosquito-associated isolates formed monophyletic groups; in particular, *S. marcescens* (Fig 4E) and *S. fonticola* clades (Fig 4F) form multiple lineages reflecting their laboratory sources. *El. anophelis* from MosAIC form a distinct mosquito-associated lineage and do not branch with the previously sequenced mosquito-derived *El. anophelis* (Fig 4G and 4H).

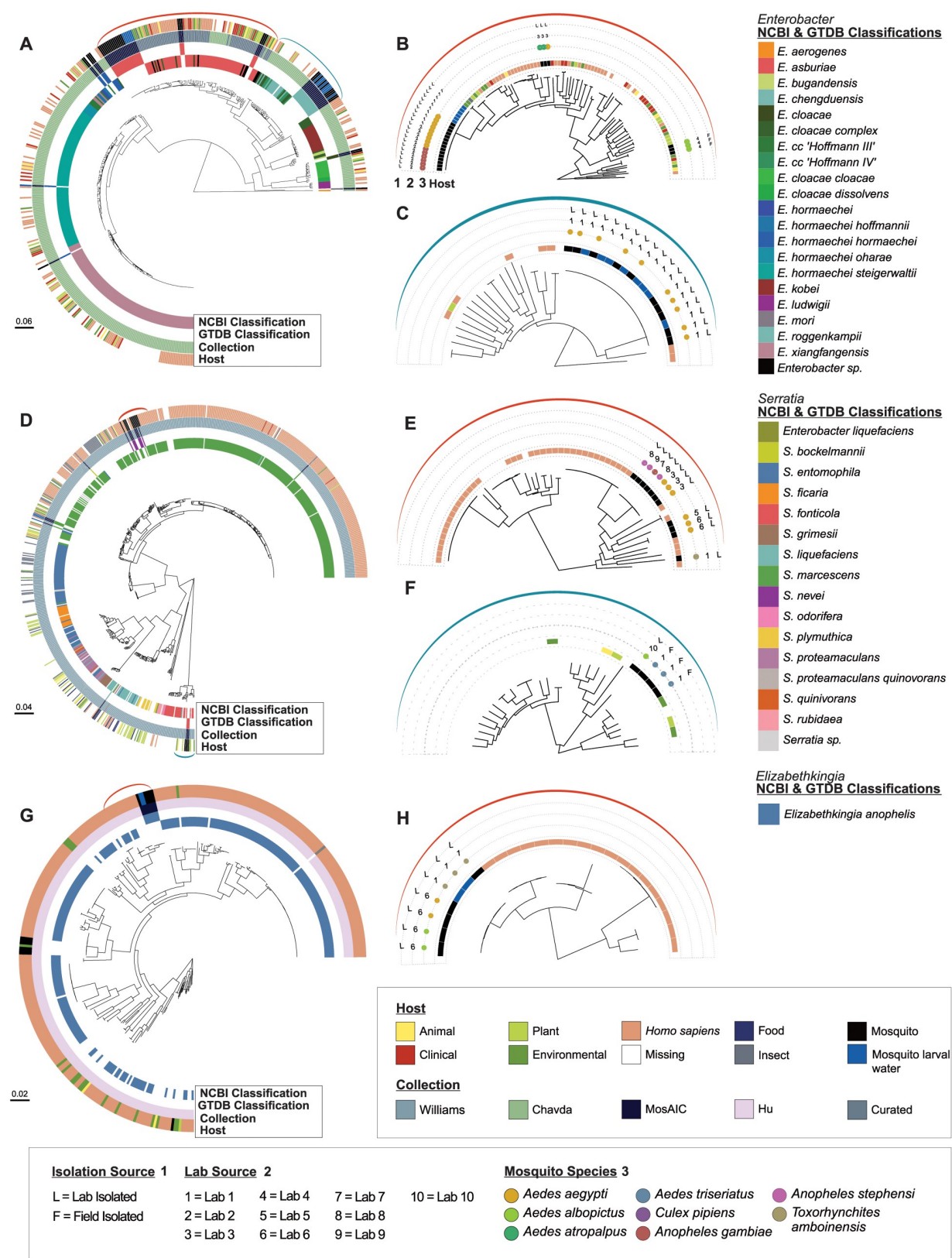

**Fig 4. Selected population structures with improved mosquito representation.** Population structures based on previously published genomic collections for (A–C) *Enterobacter* [61], (D–F) *Serratia* [62], and (G, H) *Elizabethkingia anophelis* [60], with added mosquito-derived representation from MosAIC and an additional manually curated set of publicly available *En. asburiae* genomes. Phylogenies were built using a maximum likelihood approach within IQ-TREE2 [63] and 1,000 bootstraps, using SNP-filtered core gene alignments generated with Panaroo [64] and SNP-sites [65]. The rings of each population phylogeny (A, D, G) denote, from outer to inner, host from which the sample was isolated, genomic collection from which the genome originated, GTDB classifications for the MosAIC isolates, and NCBI classifications from the original studies for the non-MosAIC isolates. Evolutionary scales are displayed on the bottom left of the figure panels. To the right of each population tree are subsets highlighting mosquito-associated lineages within a population (B, C, E, F, H), with the coloured brackets corresponding to their location within a given population tree. The rings of each subset phylogeny denote: Host (as on the population phylogenies), then 3 outer rings that show additional metadata for the mosquito-derived isolates, 1 = whether the mosquito was lab-reared (L) or field-derived (F), 2 = the laboratory group that isolated the sample, comprising some MosAIC contributors and some groups that contributed to previous studies (Lab 1 = Kerri Coon and UW-Madison Capstone in Microbiology Students, Lab 2 = Michael Povelones, Lab 3 = Michael Strand, Lab 4 = Claire Valiente Moro, Lab 5 = Douglas Brackney, Lab 6 = Eric Caragata, Lab 7 = Marcelo Jacobs-Lorena, Lab 8 = Edward Walker, Lab 9 = Sibao Wang, Lab 10 = Dong Pei), and 3 = the mosquito species the isolate was cultured from. *Enterobacter liquefaciens* within the *Serratia* phylogeny are derived from [62] and have since been reclassified as *Serratia liquefaciens*. All code and data to recreate this figure can be found at https://github.com/MosAIC-Collection/MosAIC_V1. For Fig 4A–4C, the code and data are in the folder "06b_EnterobacterPopulationStructure;" for Fig 4D–4F, they are in the folder "06a_SerratiaPopulationStructure," and for Fig 4G, 4E, and 4H, they are in the folder "06c_ElizabethkingiaPopulationStructure." GTDB, Genome Taxonomy Database; MosAIC, Mosquito-Associated Isolate Collection; SNP, single-nucleotide polymorphism.

## Genetic diversity of pangenome gene classifications of mosquito-associated bacteria

To investigate the conservation of genes within mosquito-associated bacteria, we explored the genetic diversity of *En. asburiae*, *S. marcescens*, and *El. anophelis*. The core genome of each species (gene frequency >95% across all within-species lineages) amounted to 3,109 genes in *En. asburiae*, 2,443 genes in *S. marcescens*, and 1,836 genes in *El. anophelis* (Fig 5A–5C). Pangenome gene accumulation curves had not yet plateaued for any of the 3 species indicating that the full genetic diversity of each species has not yet been realised and sequencing further isolates would likely add novel genes (i.e., all pangenomes analysed were open) (S10 Fig).

By analysing the accessory gene content and nucleotide divergence with PopPUNK, we defined 6 genome clusters that were specific to mosquito-associated isolates—3 in *En. asburiae* (S11 Fig), 2 in *S. marcescens* (S12 Fig), and 1 in *El. anophelis* (S13 Fig), which may suggest bacterial speciation or niche adaptation to the mosquito host. These lineage-specific core genes were defined as being both specific to a lineage and present at >95% frequency across that single lineage. Within *En. asburiae*, the 3 mosquito-associated lineages contained 62, 41, and 43 lineage-specific core genes; the 2 *S. marcescens* mosquito-associated lineages contained 78 and 13 lineage-specific core genes, and the single *El. anophelis* mosquito-associated lineage contained 38 lineage-specific core genes (Fig 5A–5C and S6 Table). Interestingly, within *En. asburiae* there were 2 genes conserved across 2 different mosquito-associated lineages. These corresponded to a domain of unknown function, DUF4224 (UniRef90 A0A156GGP5), shared between lineages 1 and 8, and an HNH endonuclease (UniRef90 IPI00187EE547) shared between lineages 7 and 8. Lineage-specific intermediate genes (between 15%–95% frequency) can be indicative of recently acquired or lost genes within a lineage. *En. asburiae* contained on average 33 and 2 intermediate core genes within lineages 1 and 8, respectively and the single *El. anophelis* lineage contained 20 intermediate core genes.

The genes within the mosquito-associated clusters have a variety of annotations (S14 Fig). Within *En. asburiae*, the most common annotations across all mosquito-associated isolates were "lipoprotein" (*n* = 84 annotations), "AAA family ATPase" (*n* = 66), "DNA helicase" (*n* = 42), and "phage protein" (*n* = 42). Common annotations for *S. marcescens* were "ATP-binding protein" (*n* = 16 annotations), "response regulator" (*n* = 14), and "restriction endonuclease subunit S" (*n* = 14), while common annotations for *El. anophelis* were "translocated intimin receptor (TIR)" (*n* = 21), "phage protein" (*n* = 14), and "A-deaminase" (*n* = 14). In total, we found 6 annotations shared between *En. asburiae* and *S. marcescens*, 5 annotations shared

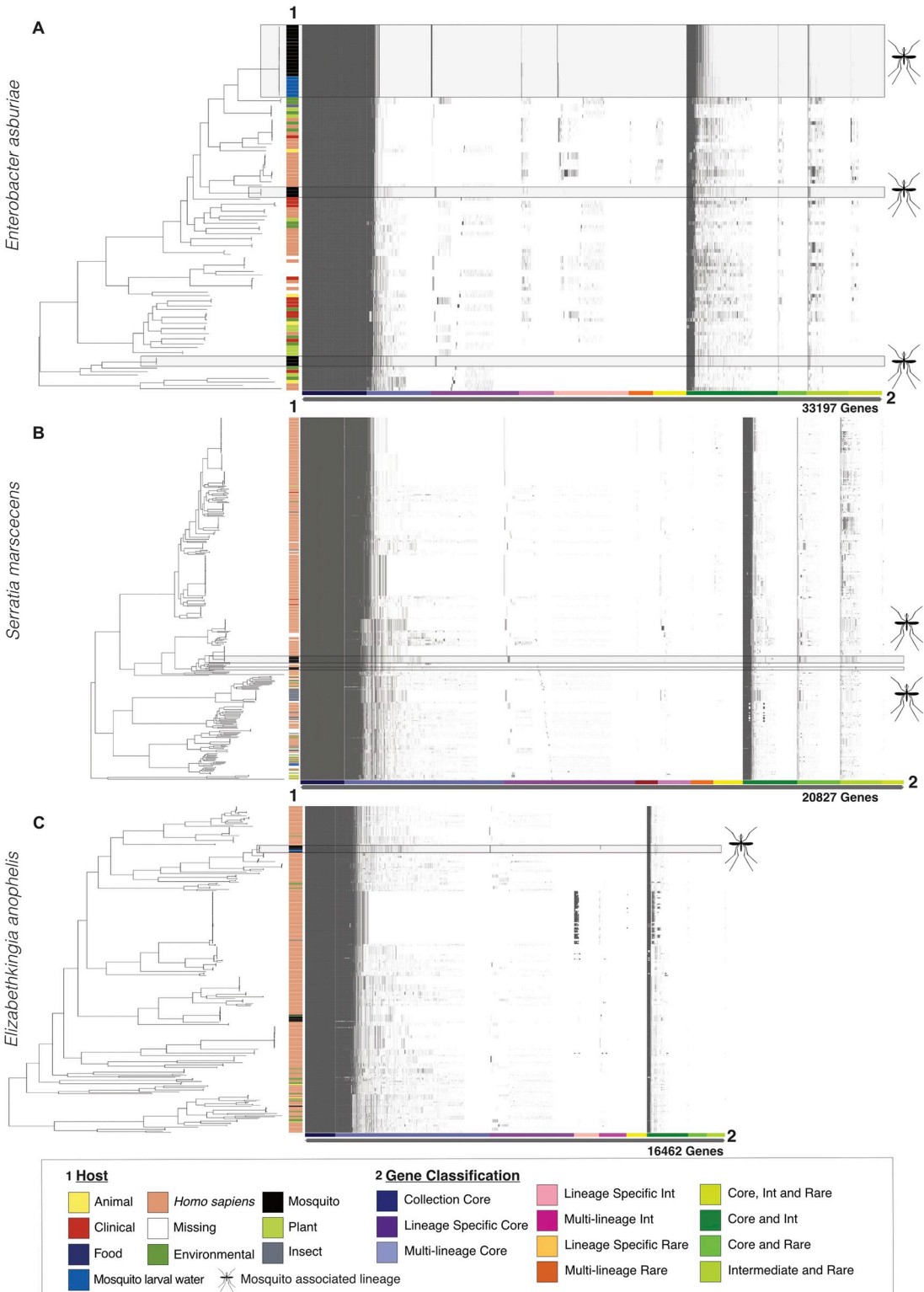

**Fig 5. Pangenomes of *Enterobacter asburiae*, *Serratia marcescens*, and *Elizabethkingia anophelis* with highlighted mosquito-associated lineages.** Panels (A–C) depict gene presence/absence within each species, generated with Panaroo [64]. Phylogenies and matrices are shaded grey to highlight mosquito-associated lineages defined by PopPUNK [66]. The *y*-axis shows the host each bacterium was isolated from, denoted as 1 Host in the figure legend. The *x*-axis shows subclassifications of the pangenome, denoted as 2 Gene Classification in the figure legend. Here, subclassification of the accessory genome was performed using the

twilight package [67]. In brief, the classification of each gene was first defined by determining their frequency within a lineage (Core, genes present in ≥95% of strains in a lineage; Int, genes present in ≥15% and ≤95% of strains; Rare, genes present in ≤15% of strains). The resulting gene classifications were then compared across each lineage using genome clusters defined with PopPUNK, which correspond to predicted lineages within the phylogeny (Collection core, genes core to the whole phylogeny; Lineage specific core, genes core to a single lineage; Multi-lineage core, genes core to ≥2 lineages). Genes defined by different classifications across lineages are given a combined class denoted by the green shading. Numbers of genes given on the *x*-axis refer to the total number of genes within each pangenome (core + accessory genes). Mosquito symbols are from https://phylopic. org. All code and data to recreate this figure can be found at https://github.com/MosAIC-Collection/MosAIC_V1. For Fig 5A the code and data are in the folder "07b_EnterobacterPangenome;" for Fig 5B, they are in the folder "07a_SerratiaPangenome," and for Fig 5C, they are in the folder "07c_ElizabethkingiaPangenome."

between *El. anophelis* and *En. asburiae*, and 1 annotation shared between *El. anophelis* and *S. marcescens* (S15 Fig). These included specific "TIR" annotations shared between *El. anophelis* and *S. marcescens*, a "HTH luxR-type domain" annotation shared between *El. anophelis* and *En. asburiae*, and a phage-associated "tail fibre assembly protein" annotation shared between *En. asburiae* and *S. marcescens* (S15 Fig).

## Discussion

The mosquito microbiome influences multiple facets of mosquito life history, including the ability of certain species to transmit human pathogens [5]. However, while current research in the mosquito microbiome field has greatly improved our understanding of the taxonomic composition of mosquito-associated bacterial communities and their broad effects on different mosquito phenotypes, our genomic understanding, and therefore the predicted function of specific community members and assemblages, remains poorly understood. Here, we present a collection of 392 mosquito-associated bacterial isolates and their genome sequences, representing the first large-scale community effort to establish a genomic isolate collection for mosquitoes.

Currently, the GTDB (Release 07-RS207) holds 36 mosquito-associated bacterial genomes from a mixture of 25 laboratory and 11 field-derived mosquitoes, across 22 species. MosAIC expands this representation to 142 species of mosquito-associated bacteria, acquired from different vector species and life stages and from individuals with different feeding statuses (i.e., blood-fed versus non-blood-fed adult females) [21,68–70]. This collection of isolates can be used to extend previous findings from amplicon-based studies [71–73] using a genotype to phenotype framework. For example, genomic investigation and functional validation of siderophore biosynthetic gene clusters [44,74] or hemolysins [75] in *Serratia* and *Elizabethkingia* may reveal why they are frequently identified post-blood meal [76–79]. Furthermore, many bacteria known to be commonly associated with mosquitoes, including *Enterobacter* [28,42], *Cedecea* [39], *Pantoea* [80,81], *Elizabethkingia* [76,82–85], *Aeromonas* [86], and *Acinetobacter* [87] are well represented in the collection. Since the collection of these isolates was not systematic and comprised more insectary-reared individuals than field-collected individuals, we did not use statistical inference to indicate differences between isolates and sources but instead highlight key observations to describe the collection. An example here are the *Acetobacteraceae* that were abundant and exclusive to non-blood-fed female and male adult mosquitoes compared to blood-fed females, which could reflect their role as sugar oxidisers [88,89]. Moreover, as with other host-associated genome collections [54,56,90,91], in compiling MosAIC we discovered 8 field-derived and 16 laboratory-derived mosquito-associated isolates (15 genera in total) with no species representation in the GTDB, highlighting the undiscovered taxonomic diversity of mosquito-associated isolates even in this opportunistic collection and using only standard bacterial growth media. Further work is required to understand their putative role in the microbiome, including characterising their morphology and growth requirements,

determining their genetic content, and testing for any specific interactions with the mosquito host and other members of the mosquito microbiome.

Mosquitoes are holometabolous insects that undergo complete metamorphosis as they develop and transition from aquatic to terrestrial habitats. During iterative phases of growth and moulting during larval stages and metamorphosis from the larval to adult stage (via a pupal stage), the microbiome is reassembled—resulting in changes in community composition across life stages [17,77,92,93]. Furthermore, the microbiome is highly variable between mosquito populations [1,18,77,94–98]. This suggests mosquitoes harbour transient bacteria, similar to butterflies and moths [99,100], rather than specialised bacteria found in social insects such as termites and honey bees that live in colonies comprised of large numbers of related individuals [101,102]. Considering the collection to date, we have observed monophyletic lineages of mosquito-associated isolates within populations of 3 taxa: *Serratia*, *Enterobacter*, and *El. anophelis*. While additional, more balanced, sampling efforts are needed to draw firm conclusions, the observation that these mosquito-associated lineages comprised isolates derived from different mosquito species and geographically distinct locations suggests potential association with the mosquito host environment. We further note that these lineages included isolates derived from laboratory cultures of the same mosquito species, although most of these cultures were established from geographically and genetically distinct field populations and have since been maintained in isolation from one another by different research groups for >30 years. Numerous studies have identified rearing environment as the dominant factor shaping microbiome composition in both larval and adult stage mosquitoes [5,45,68], even in cases where separate laboratory colonies (subcolonies) were only very recently established from the same parent colony [103]. Thus, even the observation that the mosquito-associated *En. asburiae* lineages we identified contained isolates from the same strain (Liverpool) of *Ae. aegypti* mosquitoes maintained by 2 different laboratories could still indicate a beneficial association with at least this host species. The identified genes conserved within these lineages, and the genes in *En. asburiae* common to multiple mosquito-associated lineages in particular, represent interesting candidates to explore whether any are related to mosquito-associated lifestyle. Many of the identified genes are annotated hypothetical proteins and thus are interesting candidates for further functional investigation. To expand on these promising findings, further sampling of mosquito isolates from geographically distinct, field-derived mosquitoes and functional validation of these genes is also necessary.

Mosquito-associated bacteria are acquired primarily through horizontal transmission from water to larvae, where they colonise multiple organs, primarily the midgut, with some microbiome members persisting until adulthood [21,79]. Within the mosquito, bacteria must withstand a fluctuating environment, including spatial permutations during metamorphosis [93], changes in nutrition [104,105], temperature [106], pH [107], oxidative stress [108], and competition with other members of the microbiota [23,84]. The mechanism of adaptation to both an aquatic lifestyle and colonisation in the mosquito remains poorly understood. As such, we screened MosAIC for potential host colonisation factors. This revealed multiple different virulence factor genes, including those involved in adhesion, motility, and biofilm formation. Adhesion is also likely to play a role in withstanding peristaltic movement during mosquito feeding [109], as seen in other invertebrate systems [110], and specialised motility may be required for movement through the mucosal environment of the mosquito digestive tract, similar to *Burkholderia* symbionts in the *Riptortus pedestris* bean bug [111]. It will also be important to further investigate differences between virulence factors in field and laboratory-derived mosquito isolates; for example, flagellar motility is lost in *Acetobacter* from lab-reared *Drosophila* after their close association with the fly through multiple generations, whereas they are retained in field-derived isolates originating from heterogeneous populations [112].

Virulence factor representation overall varied considerably among specific families of mosquito-associated bacteria; virulence factor genes were absent within the *Bacteroidia* due to no representation in the VFDB and a heavy skew is observed toward the *Enterobacteriaceae*. Further work to understand the expression, function, and diversity of these genes, as well as their role in free-living and mosquito-associated stages, is paramount for identifying potential candidate genes for colonisation of and interaction with the host.

The methodology used to establish MosAIC, which relied in part on crowd-sourced material, resulted in an initial collection of isolates from a diverse range of mosquito species and mosquito-associated environments. However, we acknowledge the inherent biases associated with this methodology, including the lack of systematic sampling and relatively low geographic and genetic representation within individual host species. The culturing of isolates using only standard protocols has also resulted in exclusively aerobic (obligate or facultative) bacteria in the collection, and bacteria were isolated on a limited selection of growth media, meaning a potential underrepresentation of strains requiring specific nutrients or growth conditions. Future targeted sampling will focus on obtaining isolates from underrepresented regions, in particular inviting contributions from researchers in the global south and regions with endemic mosquito-borne disease. We will also target fungi, archaea, and protozoa, which are currently lacking in the collection. Furthermore, we intend to supplement MosAIC through the inclusion of metagenomic data [113]. This will limit sampling biases through sequencing potential taxa not yet amenable to culturing [114,115] and provide a snapshot of predicted community function to supplement information from single isolates. Given the inherent taxonomic variability of mosquito microbiomes, this gives a compelling reason to address whether community function, rather than taxa, is a conserved unit of the microbiome [116] (i.e., the "song not the singer" hypothesis), as has been postulated in other micro-systems [117–120] and is comparatively well-studied in fields of macro-ecology [121]. Ultimately, the continued expansion of MosAIC, and its value for both basic and translational research to reduce the global burden of mosquito-borne diseases, will require a collaborative, community-engaged approach that leverages expertise from diverse entities within and outside of academia. Parties interested in contributing to MosAIC, accessing MosAIC resources, and/or partnering with the MosAIC team are encouraged to visit our website (https://kcoonlab.bact.wisc.edu/mosaic/).

## Conclusion

MosAIC marks a significant milestone as the first large-scale repository of physical isolates and high-quality genomic data associated to the mosquito-microbiome. Using this resource, we have begun to understand the adaptive traits of mosquito-associated bacteria by observing clusters of mosquito-associated bacterial lineages with conserved genes. Notably, we find shared genes between mosquito-associated lineages of *En. asburiae*, suggesting evolutionary convergence driven by mosquito association. To gain further insight of bacterial colonisation within this dynamic system, we emphasise probing the roles of virulence factors in these bacteria; understanding the function and expression of these genes will bring insight to their importance during mosquito development and horizontal transmission from a free-living to mosquito-associated lifestyle. Lastly, we aim to expand this collection through greater systematic sampling from field-derived, disease-endemic sources and parallel metagenomic-based sequencing efforts. This will allow us to build on our population-based observations of mosquito-adaptation to include statistical-based inferences [122] and explore community function. Addressing these questions will bring us a step closer to novel microbe-based mosquito control tools to positively impact public health and vector control efforts.

## Materials and methods

### Sample collection

**Bacterial isolations and collection curation.**   Aiming to curate a collection of bacterial isolates that encompasses the diversity of the mosquito microbiome, isolates were recovered from a diversity of mosquito species and mosquito-associated environments using a range of media and growth conditions (Fig 1 and S1 and S2 Tables). A total of 354 isolates were curated for the collection at the University of Wisconsin-Madison (UW-Madison)—80 from internal collections comprised of isolates derived from laboratory and field samples, 166 from samples provided by external contributors, and 108 from samples processed by students enrolled in MICROBIO 551 (Capstone Research Project in Microbiology, 2 credits) during the Spring 2022 semester (S1 Table). A total of 38 isolates were curated from the collection at the Liverpool School of Tropical Medicine (LSTM), from internal collections comprised of isolates derived from laboratory samples (S1 Table).

Isolates derived from existing collections at UW-Madison and LSTM were streaked from frozen glycerol stocks onto agar plates and incubated overnight. Independent bacterial colonies were then inoculated into 3 ml of liquid broth in 14-ml round-bottom tubes (Fisher Scientific, Hampton, New Hampshire, United States of America), incubated overnight with shaking at 200 rpm, and used to generate new glycerol stocks (1 ml culture + 1 ml 40% sterile glycerol) and stored in duplicate at −80˚C. Specific incubation temperatures and growth media used follow those provided in S1 Table.

Samples from external contributors were received by UW-Madison as either (i) agar stabs or liquid cultures of pure isolates inoculated directly from frozen glycerol stocks; or (ii) mixed lawns of bacteria on agar plates spread with whole mosquito homogenates. Isolates received as agar stabs or liquid cultures were immediately processed by streak-plating to ensure purity prior to long-term cryopreservation as described above. Samples received as mixed bacterial lawns were scraped into 1 ml of sterile phosphate-buffered saline (PBS, 1×), resuspended, serially diluted and plated on agar prior to incubation for up to 72 h. Single colonies were then picked for streak-plating prior to long-term cryopreservation of different morphotypes as described above. Specific incubation temperatures and growth media used follow those provided in S1 Table.

UW-Madison "Capstone in Microbiology" students isolated bacteria from samples of adult females, larvae, and larval water from internally maintained laboratory cultures of *Ae. aegypti*, *Aedes triseriatus*, *Culex pipiens*, and *Toxorhynchites amboinensis*. Adult female mosquitoes were collected via aspiration, cold anesthetised at −20˚C for 10 min, and surface-sterilised with 70% ethanol for 5 min followed by 3 rinses with sterile PBS (1×) or water. Pools of 10 adult females were then combined into a pre-sterilised 2 ml screw cap tube (Sarstedt, Newton, North Carolina, USA) containing 500 μl PBS and one 5 mm steel bead (Qiagen, Hilden, Germany) prior to bead-beating for 20 s to disrupt the cuticle. Cohorts of both sugar-fed and blood-fed adult females were collected for *Ae. aegypti*, while only sugar-fed adult females were collected for *Ae. triseriatus*, *Cx. pipiens*, and *Tx. amboinensis*. Adult female *Ae. aegypti* were blood fed using defibrinated sheep blood via an artificial membrane feeder. Larvae were collected as fourth instars, surface-sterilised and rinsed prior to homogenisation of pools of 10 larvae in 1.7 ml snap-cap tubes (Corning, Corning, New York, USA) containing 500 μl sterile PBS (1×), using a sterile plastic pestle (Bel-Art Products, Wayne, New Jersey, USA). Larval water from trays containing fourth instar larvae was collected in 50 ml screw-cap tubes (Thermo Fisher Scientific, Waltham, Massachusetts, USA), centrifuged for 5 min at 8,000 × *g*, and resuspended in 2 ml PBS. Resulting homogenates were mixed with an equal volume of sterile PBS (1×) containing 40% glycerol and stored at −80˚C prior to student isolations.

Isolations were performed by diluting material from frozen stocks to extinction in YTG broth (up to $10^{-5}$ in 96-well plates) and observing cultures for growth for 21 days at 30°C. Wells showing growth were then struck onto YTG agar for isolation as described above.

**DNA extraction and sequencing.** DNA was extracted from each sample using the DNeasy Blood and Tissue kit (Qiagen, Hilden, Germany), following manufacturer's conditions, with those samples extracted at UW-Madison (S1 Table) being subjected to the recommended pretreatment step for isolation of gram-positive bacteria, and the samples prepared at LSTM being subjected to the recommended pretreatment step for gram-negative bacteria. DNA extracted at LSTM was quantified using a Qubit fluorometer (Thermo Fisher Scientific, Waltham, Massachusetts, USA) and used to generate sequencing libraries using the NEBNext Ultra II FS Library Prep kit (New England Biolabs, Ipswich, Massachusetts, USA), followed by sequencing on the Illumina MiSeq platform to generate 250 bp paired end reads. DNA extracted at UW-Madison was quantified using a Quantas fluorometer (Promega, Madison, Wisconsin, USA) and sent to the UW-Madison Biotechnology Center Next Generation Core facility for library preparation using the Celero EZ DNA-Seq Library Prep kit (Tecan, Männedorf, Switzerland), followed by sequencing on the Illumina NovaSeq6000 platform to generate 150 bp paired end reads.

## Bioinformatics

**Quality control, assembly, and filtering.** An overview of the analysis pipeline is shown in S16 Fig. Quality of raw reads was assessed using FastQC v0.11.9 [123] by examining average quality scores across read sequences to remove samples with average phred scores <20. Assemblies were generated with Shovill v1.1.0 [124] (https://github.com/tseemann/shovill) at default settings, which uses the assembler Spades v3.14 [125] at its core with pre and post-processing steps to refine the final assemblies. The resulting contiguous sequences were assessed for quality using both QUAST v5.2.0 [126] and CheckM v1.2.2 [127].

**Genome classification and phylogenetic reconstruction.** The resulting genome assemblies were taxonomically classified against the GTDB (Release 07-RS207) [128] using the taxonomic classifier GTDB-Tk v2.1.1 [129]. To reconstruct a phylogeny encompassing the diversity of the collection, we first acquired species level representatives using dRep v3.0.0 [130]. Within dRep, clustering was conducted with a primary threshold of 90% using MASH v1.1.1 [131], followed by a secondary clustering step using a 95% ANI threshold with FastANI v1.33 [132]. Ribosomal RNA sequences within each of the species level representatives were predicted using barrnap v0.9 (https://github.com/tseemann/barrnap). Barrnap [133] predicts rRNA sequences using HMMER3 v3.3.2 [134], with models built using the SILVA and Rfam databases. Predicted 16S rRNA sequences were then extracted and aligned using MUSCLE v5.1.0 [135] with default settings. The resulting alignment was trimmed using TrimAL v1.4.1 [136], with a heuristic method optimised for maximum likelihood tree reconstruction (—automated1). The refined alignment was used to reconstruct the phylogenetic tree using IQ-TREE2 v2.0.6 [63] with 1,000 ultrafast bootstraps and a TIM3+F+R6 model predicted using Model-Finder [42]. The tree was displayed using a 16S rRNA sequence from the *Synergistales* order, *Synergistes jonesii* (GCF_000712295.1), as a root [137]. The resulting phylogeny was annotated using the GTDB-Tk taxonomic classifications and the number of isolates per species.

**Virulence factors.** Virulence factors were predicted using ABRicate [138,139] v1.0.1 (https://github.com/tseemann/ABRicate). The default virulence factor database included in ABRicate was replaced with DNA sequences from the full data set to include all genes related to known and predicted virulence factors [140], (http://www.mgc.ac.cn/VFs/download.html). All gene hits were combined into a single report using the—summary command within

ABRicate. Metadata of this database was also examined to describe patterns and biases related to the bacterial origin of these genes (S6 Fig).

**MosAIC isolates in phylogenetic context.** We explored the phylogenetic relationships of 3 mosquito derived taxa in the collection: *Enterobacter*, *Serratia*, and *El. anophelis*. These have defined population structures [60–62], are well represented in the collection, and are common mosquito symbionts [21,34]. Genomes as described in previous studies [60–62] were retrieved from GenBank (S5 Table). Given the limited number of *Enterobacter asburiae* genomes in [61], we included additional *En. asburiae* genomes from the GTDB, filtered by CheckM >98% completeness <5% contamination and existence in a published study (S5 Table). Both MosAIC and published genomes were (re)annotated with Bakta v1.7 [141] to ensure consistency. For each of the 3 data sets, Panaroo v1.3.2 [64] was used to retrieve core gene alignments. Here, all genes present in at least 95% of samples were aligned using MAFFT with clean mode set to moderate, a threshold of 70% identity, and removal of invalid genes such as those with a premature stop codon (—threshold 0.95 –clean-mode moderate -f 0.7 -a core–aligner mafft–core_threshold 0.95 –remove-invalid-gene). Variant sites containing single-nucleotide polymorphisms (SNPs) were extracted from the core gene alignments using SNP-Sites v2.5.1 [65], allowing for gaps, and the resulting core SNP alignments were used to reconstruct the phylogenies, placing MosAIC isolates in the larger population structures. Each tree was built using 1,000 bootstraps with IQ-TREE2 v2.0.6 [63] with the highest scoring models determined using ModelFinder [142]; TIM2e+ASC+R4 for *Serratia*, consistent with [62], GTR+F+I+I+R4 for *Enterobacter* and SYM+R3 for *Elizabethkingia*. The *Serratia* phylogenetic tree display was rooted between *Serratia fonticola* and the remaining *Serratia* species, as defined by the original population structure [62]; *Enterobacter* and *Elizabethkingia* displays were rooted using the outgroups *Klebsiella aerogenes* and *Chryseobacterium bovis*, respectively.

**Pangenome analyses.** Clusters for the 3 data sets were defined using PopPUNK v2.6.0 [66], which uses pairwise nucleotide k-mer comparisons to determine shared sequence and gene content and therefore clusters assemblies according to core and accessory gene content. For *En. asburiae* and *El. anophelis*, a Bayesian Gaussian mixture model was used to build a network to define clusters. For *S. marcescens*, a HDBSCAN model was used to reflect the larger number of isolates in the phylogeny as recommended by the authors [66]. The subsequent refinement step was run using "—fit-model refine" for all predicted clusters to improve overall network scores.

Using PopPUNK-defined lineages and the Panaroo gene presence-absence matrix, we performed population-structure aware gene classifications using the *Twilight* package [67]. We set the minimum lineage cluster size to one to account for singleton lineages and core and rare thresholds were set to 0.95 and 0.15, respectively. Gene accumulation curves were generated using the specaccum function within the *vegan* package [143].

All data analysis was conducted in *R* v4.2.2 [144] using the following packages: *tidyverse* v1.3.2 [145], *ggtree* v3.16 [146], *ggtreeExtra* v3.17 [147], *treeio* v3.17 [148], *APE* v5.71 [149], *janitor* (https://github.com/sfirke/janitor), and *phytools* v1.5 [150].

## Supporting information

**S1 Table. Curated file containing all isolate-associated metadata, select genome quality-assurance metrics, GTDB classifications, and genome accession numbers.** unk = unknown, NA = not applicable.
(XLSX)

**S2 Table. Summary table of mosquito–bacterial origin.** Metadata category names and definitions follow those presented in S1 Table. NA = not applicable.
(DOCX)

**S3 Table. Complete taxonomic classification of all MosAIC isolates, assigned via GTDB-Tk.**
(XLSX)

**S4 Table. Virulence factor genes identified in MosAIC isolates using VFDB.**
(XLSX)

**S5 Table. Twilight gene classifications for *En. asburiae*, *El. anophelis*, and *S. marcescens* isolates from MosAIC.**
(XLSX)

**S6 Table. Metadata for previously sequenced (i.e., external to MosAIC) *Enterobacter*, *Serratia*, and *Elizabethkingia* genomes retrieved from GenBank and manually curated from the GTDB.**
(XLSX)

**S1 File. Consortium author list.**
(DOCX)

**S2 File. README file outlining how repository code/data can be used to reproduce all figures in the manuscript.**
(DOCX)

**S1 Fig. Distribution of raw sequencing counts.** Histogram showing the size distribution of raw sequencing reads used to assemble MosAIC genomes. The *x*-axis shows the number of reads per isolate, while the *y*-axis shows the number of isolates with a specific read count. All code and data to recreate this figure can be found at https://github.com/MosAIC-Collection/MosAIC_V1 in folder "01_GenomeQC."
(TIFF)

**S2 Fig. Correlation between number of predicted genes and genome size across MosAIC isolates.** Scatterplot showing the relationship between genome size (*x*-axis) and number of predicted genes (*y*-axis) for each isolate. Each point represents a high-quality genome assembly (>CheckM completeness 98%, <CheckM contamination 5%, >10× read coverage). Line fitted using a linear model in R. Mbp = megabase pairs. All code and data to recreate this figure can be found at https://github.com/MosAIC-Collection/MosAIC_V1 in folder "01_GenomeQC."
(TIFF)

**S3 Fig. Twenty-five isolates within MosAIC share <95% ANI to a reference genome in the GTDB.** The *x*-axis of the bar chart shows the number of isolates assigned to a reference genome with a given genus assigned taxonomy in the GTDB (*y*-axis). "JAATFS01" is a strain identifier placeholder used by GTDB-Tk when no binomially named representative genome is present in the GTDB. All code and data to recreate this figure can be found at https://github.com/MosAIC-Collection/MosAIC_V1 in folder "02_GTDB_Drep_Summary."
(TIFF)

**S4 Fig. Bacterial family distribution of MosAIC isolates assigned to different metadata categories.** The *x*-axis of each bar chart shows the number of isolates assigned to a reference genome with a given family assigned taxonomy in the GTDB (*y*-axis). Charts are faceted by metadata category as follows: (A) female_feeding_status, (B) lab_field_derived, (C)

mosquito_sex, and (D) mosquito_tissue. Metadata category names and definitions follow those presented in Table S1. All code and data to recreate this figure can be found at https://github.com/MosAIC-Collection/MosAIC_V1 in folder "12_Metadata_Exploration."
(PDF)

**S5 Fig. Bacterial genus distribution of MosAIC isolates assigned to different metadata categories.** The *x*-axis of each bar chart shows the number of isolates assigned to a reference genome with a given genus assigned taxonomy in the GTDB (*y*-axis). Charts are faceted by metadata category as follows: (A) lab_field_derived, and (B) mosquito_sex. Metadata category names and definitions follow those presented in S1 Table. Only isolates assigned to genera within the *Enterobacteriaceae* were included in the analysis. All code and data to recreate this figure can be found at https://github.com/MosAIC-Collection/MosAIC_V1 in folder "12_Metadata_Exploration."
(PDF)

**S6 Fig. Taxonomic representation across the VFDB.** Plots demonstrate the bias in database composition, which shows a strong skew towards members of the Gammaproteobacteria. Plots are faceted by bacterial class, with the *x*-axis of each chart showing the number of isolates assigned to a given genus on the *y*-axis in which at least 1 virulence factor gene was identified. All code and data to recreate this figure can be found at https://github.com/MosAIC-Collection/MosAIC_V1 in folder "05_Virulence_Factor_Analysis."
(TIFF)

**S7 Fig. *Enterobacter* Phylogeny with tip labels.** Tip labels of *Enterobacter* are shown at each node. All code and data to recreate this figure can be found at https://github.com/MosAIC-Collection/MosAIC_V1 in folder "06b_EnterobacterPopulationStructure."
(EPS)

**S8 Fig. *Serratia* Phylogeny with tip labels.** Tip labels of *Serratia* are shown at each node. All code and data to recreate this figure can be found at https://github.com/MosAIC-Collection/MosAIC_V1 in folder "06a_SerratiaPopulationStructure."
(EPS)

**S9 Fig. *Elizabethkingia anophelis* Phylogeny with tip labels.** Tip labels of *Elizabethkingia anophelis* shown at each node. Green bar denotes mosquito-associated samples from MosAIC. All code and data to recreate this figure can be found at https://github.com/MosAIC-Collection/MosAIC_V1 in folder "06c_ElizabethkingiaPopulationStructure."
(EPS)

**S10 Fig. Gene accumulation curve.** Pangenome gene accumulation curve for *En. asburiae*, *El. anophelis*, and *S. marcescens* isolates from MosAIC. All code and data to recreate this figure can be found at https://github.com/MosAIC-Collection/MosAIC_V1 in folder "08_GeneAccumulationCurve."
(TIFF)

**S11 Fig. *Enterobacter asburiae* phylogeny overlaid with PopPUNK defined genome clusters.** Tips denote PopPUNK cluster. Green highlight denotes mosquito-associated lineages containing MosAIC isolates. All code and data to recreate this figure can be found at https://github.com/MosAIC-Collection/MosAIC_V1 in folder "10_VisPopPUNKClusters."
(TIFF)

**S12 Fig. *Serratia marcescens* phylogeny overlaid with PopPUNK defined genome clusters.** Tips denote PopPUNK cluster. Green highlight denotes mosquito-associated lineages

containing MosAIC isolates. All code and data to recreate this figure can be found at https://github.com/MosAIC-Collection/MosAIC_V1 in folder "10_VisPopPUNKClusters."
(EPS)

**S13 Fig. *Elizabethkingia anophelis* phylogeny overlaid with PopPUNK defined genome clusters.** Tips denote PopPUNK cluster. Green highlight denotes mosquito-associated lineages containing MosAIC isolates. All code and data to recreate this figure can be found at https://github.com/MosAIC-Collection/MosAIC_V1 in folder "10_VisPopPUNKClusters."
(TIFF)

**S14 Fig. Annotations of lineage-specific core genes identified from mosquito-associated lineages.** Panels summarise annotations for one of 3 focal species (*Elizabethkingia anophelis*, left; *Serratia marcescens*, centre; or *Enterobacter asburiae*, right), with the *x*-axis of each panel denoting the internal identifier of individual isolates assigned to each species as presented in S1 Table. Tiles denote the number of identified annotations corresponding to a given functional category on the *y*-axis, following a gradient from dark blue (few) to light blue (many). White tiles denote categories for which zero annotations were identified in each isolate. All code and data to recreate this figure can be found at https://github.com/MosAIC-Collection/MosAIC_V1 in folder "11_LineageCoreGeneAnalysis."
(TIFF)

**S15 Fig. Shared-core gene annotations among mosquito-associated lineages.** Tiles denote the number of identified annotations corresponding to a given functional category on the *y*-axis that were shared between a given species pair on the *x*-axis, following a gradient from dark blue (few) to light blue (many). White tiles denote categories for which zero shared annotations were identified in each species pair. All code and data to recreate this figure can be found at https://github.com/MosAIC-Collection/MosAIC_V1 in folder "11_LineageCoreGeneAnalysis."
(TIFF)

**S16 Fig. Analysis workflow.** Flowchart describing (A) the assembly of MosAIC genomes and (B) population genomic analyses.
(TIFF)

## Acknowledgments

We thank Lyric Bartholomay and Kathy Vaccaro (UW-Madison Department of Pathobiological Sciences) for assistance with maintaining the *Ae. aegypti*, *Ae. triseriatus*, *Cx*. pipiens, and *Tx. amboinensis* colonies used to generate material for isolations made by the 2022 UW-Madison Capstone in Microbiology Students included in this study. We also thank Aldo Arellano, Andrew Sommer, and Serena Zhao for assistance with maintenance of internal collections used by personnel in Kerri Coon's laboratory in the UW-Madison Department of Bacteriology. Additional funding support for the Capstone in Microbiology course developed as part of this project was provided by the UW-Madison Department of Bacteriology. Additional funding support for whole genome sequencing was provided by the Wisconsin Alumni Research Foundation (to KLC). We thank Martin Donnelly for equipment access and Sanjay C Nagi for technical support.

## Author Contributions

**Conceptualization:** Kerri L. Coon, Eva Heinz, Grant L. Hughes.

**Data curation:** Laura E. Brettell, Kerri L. Coon, Aidan Foo, Eva Heinz, Holly L. Nichols.

**Formal analysis:** Laura E. Brettell, Aidan Foo, Holly L. Nichols.

**Funding acquisition:** Kerri L. Coon, Eva Heinz, Grant L. Hughes.

**Investigation:** Laura E. Brettell, Vishaal Dhokiya, Aidan Foo, Ananya F. Hoque, Jessica A. Lysne, Miguel Medina Muñoz, Holly L. Nichols, 2022 UW-Madison Capstone in Microbiology Students.

**Methodology:** Laura E. Brettell, Kerri L. Coon, Aidan Foo, Eva Heinz, Holly L. Nichols.

**Project administration:** Kerri L. Coon, Eva Heinz, Grant L. Hughes.

**Resources:** Doug E. Brackney, Eric P. Caragata, Kerri L. Coon, Eva Heinz, Grant L. Hughes, Michael L. Hutchinson, Marcelo Jacobs-Lorena, David J. Lampe, Edwige Martin, Claire Valiente Moro, Timothy D. Paustian, Michael Povelones, Michelle R. Rondon, Sarah M. Short, Blaire Steven, Jiannong Xu.

**Software:** Laura E. Brettell, Kerri L. Coon, Aidan Foo, Eva Heinz, Holly L. Nichols.

**Supervision:** Laura E. Brettell, Kerri L. Coon, Eva Heinz, Grant L. Hughes, Holly L. Nichols, Timothy D. Paustian, Michelle R. Rondon.

**Validation:** Laura E. Brettell, Kerri L. Coon, Vishaal Dhokiya, Aidan Foo, Eva Heinz, Ananya F. Hoque, Jessica A. Lysne, Miguel Medina Muñoz, Holly L. Nichols.

**Visualization:** Laura E. Brettell, Kerri L. Coon, Aidan Foo, Eva Heinz, Holly L. Nichols.

**Writing – original draft:** Laura E. Brettell, Kerri L. Coon, Aidan Foo, Eva Heinz, Grant L. Hughes, Holly L. Nichols.

**Writing – review & editing:** Laura E. Brettell, Kerri L. Coon, Vishaal Dhokiya, Aidan Foo, Eva Heinz, Ananya F. Hoque, Grant L. Hughes, Jessica A. Lysne, Miguel Medina Muñoz, Holly L. Nichols.

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
