## [Editor Report · Decision Letter 0]

9 Nov 2023

Dear Dr. Brettell, 

Thank you for submitting your manuscript entitled "Establishment and comparative genomics of a collection of mosquito-associated bacterial isolates - MosAIC (Mosquito-Associated Isolate Collection)" for consideration as a Methods and Resources by PLOS Biology.

Your manuscript has now been evaluated by the PLOS Biology editorial staff and I am writing to let you know that we would like to send your submission out for external peer review.

Once your full submission is complete, your paper will undergo a series of checks in preparation for peer review. After your manuscript has passed the checks it will be sent out for review. To provide the metadata for your submission, please Login to Editorial Manager (https://www.editorialmanager.com/pbiology) within two working days, i.e. by Nov 11 2023 11:59PM.

Kind regards,

Paula

---

Senior Editor

PLOS Biology

---

## [Decision Letter · Decision Letter 1]

26 Jan 2024

Dear Dr Brettell,

Apologies for the back-to-back emails, but we were actually able to finalize our decision for your PLOS Biology manuscript, "Establishment and comparative genomics of a collection of mosquito-associated bacterial isolates - MosAIC (Mosquito-Associated Isolate Collection)", shortly after I sent you our update email. Your manuscript has now been evaluated by the PLOS Biology editors, an Academic Editor with relevant expertise, and by several independent reviewers. I am writing to let you know that, in light of the reviews, which you will find at the end of this email, we would like to invite you to revise the work to thoroughly address the reviewers' reports.

As you will see below, the reviewers agree that your study has the potential to be a very useful resource for the community. However, they raise a number of important concerns and suggestions, and we think that these will need to be thoroughly addressed before we can consider your study for publication, as doing so will enhance the resource value of this work. We would like to emphasize the need to make the isolates and data collected here easily accessible to researchers, via deposition to publicly available repositories. We also think that the manuscript will need to be textually revised, as the reviewers suggest, focusing more on the resource aspect.

Given the extent of revision needed, we cannot make a decision about publication until we have seen the revised manuscript and your response to the reviewers' comments. Your revised manuscript is likely to be sent for further evaluation by all or a subset of the reviewers.

**IMPORTANT - SUBMITTING YOUR REVISION**

*Re-submission Checklist*

*Published Peer Review*

*PLOS Data Policy*

*Blot and Gel Data Policy*

Thank you again for your submission to our journal and I apologize again for the protracted review process. We hope that our process has been constructive thus far, and we welcome your feedback at any time. Please don't hesitate to contact us if you have any questions or comments.

Sincerely,

Luke

Lucas Smith, Ph.D.

Senior Editor

PLOS Biology

lsmith@plos.org

on behalf of 

Richard Hodge, Ph.D.

Senior Editor

PLOS Biology

rhodge@plos.org

REVIEWS:

Reviewer #1: Description of Study:

This manuscript describes a collection of bacterial isolates obtained from field and laboratory environments containing mosquitoes. This MosAIC collection contains isolates from several species of mosquitoes and from different geographic localities. In total it has 392 isolates cryopreserved in -80C glycerol stocks with plans to collect more and to diversify in collecting fungi and microbial eukaryotes from mosquito habitats. Additionally, bacterial isolate genomes were individually sequenced with Illumina technology and the manuscript describes the taxonomic diversity, genomic completeness, genetic content focused on virulence factors & pangenome content, and the phylogenetic placement of mosquito-associated isolates among closely related genomes (publicly available) for 3 highly-sampled genera.

 Overall, this is a well-written paper that describes a large collection of bacteria from mosquito-associated environments/guts. This initiative set out to collect isolates in a non-systematic way and therefore could not compare isolates or sampling locations in a statistical framework to draw more information about the associations of different microbial species/genera. This is a resource that has the potential to be used by numerous research groups; however, I feel that before this manuscript can be published there are several things that need to be done. I detail minor and major issues below.

Major Issues essential for publication:

This manuscript is presented as a Resource for the scientific field of host-microbe interactions (specifically mosquito researchers); this is both the physical collection of isolates and the genomic assemblies. This is specifically stated in the Abstract (Lines 56-58). Furthermore, as per the "Data Availability" Section "simply stating 'data available on request from the author' is not acceptable". In reading this manuscript the authors state that the isolates are not yet available in a culture collection (Lines 552-554), and that genome assemblies are not available - only raw data (Lines 653-654). Before publication I believe that 1) the isolates must be deposited and accessioned into a culture collection so that scientists can make requests, and 2) genome sequences for each isolate must be accessioned into GenBank or a similar database to make them public. These steps are important for MosAIC to become a resource that is utilized by the broader field and not to be exclusive to the authors of this manuscript. Without these resources (strains and genomes) being made publicly available, this work does not meet the PLoS Biology threshold for "Data Availability". 

 I would note that Table S1 lists the genome assembly accessions (Column AW) as "TBC" (I assume an abbreviation of [To be completed]); and that isolates are in the process of submission to a culture collection (Lines: 552-554).This suggests that the authors plan to submit the genomes and isolates - I ask that these tasks are completed before publication.

Citations from above:

Lines: 56-58 of the Abstract: "we generated a bacterial Mosquito-Associated Isolate Collection (MosAIC) consisting of 392 bacterial isolates with extensive metadata and high-quality draft genome assemblies that are publicly available for use by the scientific community."

Lines 552-554: "Physical isolates for all 392 samples that passed our genome quality thresholds are in the process of being deposited for long-term preservation and public access via culture repository and distribution centers based in the U.S. and Europe. Individual isolates are also available upon request."

Lines 653-654: "Raw Illumina reads are available in the NCBI Sequence Read Archive (https://www.ncbi.nlm.nih.gov/sra) under BioProject ID PRJNA1023190."

Minor Issues:

Several isolates in MosAIC are very closely related (often monophyletic) even as they were collected from different laboratories in geographically different locations. This raises the question of "where did these isolates come from?". The manuscript comments on this a bit, but one aspect that wasn't addressed is the relationship of mosquito colonies among labs - are some of them passed mentor-to-mentee, or between collaborators? This provenance is likely a very important aspect of collecting truly independent samples of mosquito-associated microbial communities from laboratory colonies. Related to this topic is the question "do labs use the same diet for feeding mosquitoes?". The dietary products introduced into rearing water is an obvious source of microbial communities - do the labs use the same diets/providers?

 I am not sure if this information is available, but I believe that addressing these issues in the manuscript would be useful for the field of mosquito-microbe interactions. This could be information in table format (diets used by labs) or in a new figure of mosquito colony origins and links between lab colonies. At a minimum, these topics should be mentioned in the manuscript text.

While the genomic resources associated with MosAIC represent a considerable amount of data, I am unsure about how much use it will receive in future research. This is because the authors utilized Illumina technology which does not produce complete genome sequences. With current long-read options available, bacterial genomes are easily sequenced to completion (whole chromosome(s) + plasmids). These complete genomes are very important for comparative genomics because researchers can be confident that they have all genes in that genome, and it allows for work on gene order (synteny) which is important for HGT analysis. Examining Table S1 (column AC) I found that none of the MosAIC genomes are in a single contig (min = 6; max = 1382; average = 114 contigs) and that approximately half of the isolates are in >80 contigs. 

Reviewer #2: In this manuscript, Foo et al introduce a library of bacterial isolates originating from mosquitoes and their environment, called MosAIC. They sequenced the whole genomes of their 392 isolates, spanning 142 bacterial species. These bacteria belong to classes and families known to be highly represented in mosquitoes. A large proportion of the isolates were isolated from laboratory samples, and some also derived from field samples.

This manuscript is prepared with care, hence I don't have any comments on the introduction, discussion and methods. As it contains a lot of information, I hope that my following comments will help the authors improve the readability in the results.

Figure 2A would benefit from a clearer separation between bacterial classes on the chart. "Points at the tip of bars": do the authors refer to the stars? If so, do they correspond to Figure 2F? In this case, the authors may clarify (for instance, using numbers instead of the stars)

Figure 2B: can the authors increase the size of the dots?

Figure 2D: what is N50?

In the text referring to Fig S4A, the authors could more clearly indicate that examples are discussed (several other families are represented in only non-blood fed or only blood fed samples). If I am not mistaken, figure items S4C and S4D are not described in the text.

Line 253: I don't think that the beginning of the sentence matches well the end. The authors may mean "between bacterial families", or "within bacterial classes"...

Line 260: Not sure "Fig S6" corresponds to that sentence. Can the authors clarify?

Line 274: not Fig4B => Fig 4D, E, or F?

Figure 4 is not easy to get at first sight. Even though I acknowledge that this is due to a large amount of provided information, I would suggest:

- to make a visual link between the coloured bows in A and in B / C so that the subsets appear more obviously (and the same in D => E/F and in G => H)

- to use "F" and "L" instead of the circles and stars for field and lab, and 1-9 for the lab references. Otherwise the reader who wants to know from which lab comes an isolate needs to go from the shape to the number, and from the number to the lab name, that's complicated. One step could be easily removed.

- clearly separate Mosquito species 3 from Host and collection in the legends, and clarify what is in A,D, G and what is in the other figure items.

Line 331: the numbers of genes don't correspond to those mentioned in Fig A-C. Please clarify in the text and/or figure.

Line 356: two or three in En. Asburiae? Only two appear in fig S11

Fig S14: it would be clearer to keep the same order in the three species as in the rest of the manuscript (Enterobacter => Serratia => Elizabethkingia). And wouldn't it make sense to sub classify the isolates of each species into their clusters?

Reviewer #3: Dear Authors and Editorial Staff at PLoS Biology,

This is a potentially outstanding contribution to microbiome science. I applaud the selfless and progressive efforts of these scientists, colleagues who share an interest in the mosquito microbiome, to create an open-science resource that will be extremely valuable to our larger community. Overall, I am highly supportive of their efforts and I would like to see this work published quickly. Moreover, I am excited to contribute to and utilize this emerging resource when it is fully-launched.

I do have some critical comments that I would like to see addressed before this is ultimately accepted for publication. Many of my most critical comments appear to be well-understood by the authors, particularly the limitation of the dataset imposed by the sampling design. My major suggestion is that the authors pivot in the discussion to discuss key features of this proposed resource that must be developed, rather than discussing weakly-substantiated scientific observations that at this stage remain relatively dubious, particularly those that generalize features of mosquito symbiont genomes and function. 

Nevertheless, I encourage the editor to work with these authors on this important manuscript. I am optimistic that a resubmission after some thought on my comments below could be published at PLoS Biology.

Overall comments.

This dataset is highly flawed, specifically in its ability to generalize features of the bacterial genome shared among mosquito symbionts. Typical of haphazard collection designs, the dataset is unbalanced with respect to host and environmental substrates, host taxa (mosquito or non-mosquito), mosquito species, field or laboratory populations, host life stage, geography, and host sex. A review of the supplemental files reveals that accounting for this imbalance statistically would be extremely difficult and underpowered. Though the authors share my trepidation in the interpretation of this dataset (e.g. they rightfully resist significance testing and are very clear about the datasets limitations on identifying general genomic features of mosquito-associated symbionts), they utilize a substantial portion of the results and discussion to highlight these comparative patterns found with this dataset. At best, these remain untested hypotheses based on observations. While these observations form the foundation for much of biological inquiring, it is too early in that process for publication based solely on these merits alone. 

With that being said, this is a tremendous resource and a superb example of collaborative science to solve a problem limiting progress in an important and popular field. It is very worthy of publication on these merits alone, and I am thoroughly impressed with the efforts of these scientists to coordinate a resource like this. As a "Methods and Resources" contribution to PLOS Biology, I would like to see the discussion about the growth of this resource emphasized. The authors are the best positioned to guide the community on fundamental and open-ended questions on the functionality of MosAIC, how it can be used, and how it can be improved. Specifically, I pose the following questions and ask the authors to clarify these in a resubmission.

1. Ideally, what is the detailed plan to make these samples available to other researchers? How will these isolates be maintained? How will they be distributed? Who will make the decision on this access and distribution?

2. How are issues with data sovereignty integrated into the expansion plan? For instance, who owns the isolates? If IP is generated, where do the resulting resources go? Are communities from areas where the isolates were sourced integrated into these plans?

3. What data are needed to expand the scientific or hypothesis-testing value of the dataset? If the authors would see an influx of isolate submission, where would they hope these isolates are coming from? What dimensions of the metadata remain inadequately sampled and limit our ability to use these data to uncover real patterns among in the genomes and functions of mosquito-associated microbial symbionts? 

In summary, I am highly supportive of this work. In a resubmission, I'd like to see a deemphasis on (but not necessarily a total abandonment of) the observational analysis of the existing dataset, in favor of more discussion on how this resource should be/will be implemented and grown. 

Specific comments for consideration.

Line 106- The use of "we" might be reconsidered as the author list of the cited manuscripts overlaps but does not exhaust the authors of this contribution.

Line 140- It would be helpful to add the number of bacteria isolates obtained from hosts generated in a lab-setting versus those sampled from nature in the main text.

Line 148/Figure 1. While visually interesting, I find this figure difficult to interpret. It would seem like this was created in order to aid the reader that might be interested, for instance, in the number of isolates from "Ae. albopictus female mosquitoes collected from the lab that were blood-fed". However, this is challenging to determine qualitatively from the figure and impossible to determine quantitatively. It seems a boring table might be more effective here. Consider replacing the figure with a table. 

Line 169- Please define "ANI" as Average Nucleotide Identity here in the main text, as this is the first place it is introduced to the reader.

Line 206- Given the sampling regime, I wonder if this statement has enough support to actually state. I fear it could have a misleading implication.

Line 246. Can you clarify "uniquely present genes" (i.e., unique to what? Class? Family? etc)?

Line 271. Which environmental niches are represented among these isolates?

Line 301-318. When one states "laboratory", is that meant to state the researchers that isolated the bacteria regardless of its source in a host from nature or a lab colony, or is it derived from a specific lab colony?

Line 406. "The microbiome is reassembled-resulting in altering composition across life stages". The use of "altering" in this sentence is awkward. Consider revision.

Line 422-429. Might this change with more balanced sampling design? How do we know that these patterns are not the resent of bias sampling or hierarchies/sample dependencies in dataset? 

Line 478. If this is to become a community resource, this will be the role of the whole scientific community interested in mosquito symbionts. I would recommend using more of this manuscript (specifically, the discussion) to lay out a specific plan for this. These authors know the limitation of their data better than anyone else. I believe they should set the narrative and call-to-action to shore up the current inadequacies of this dataset.

Line 482. I think it's worth mentioning that this is a well-established and reasonably supported hypothesis in "macro-organismal" community ecology. The support is particularly strong in plant communities and aquatic pool communities. Functional traits tend to underlie more predictable patterns of community assembly (i.e. determinism). In general, microbiome science (particularly those with associated with mosquito vectors) should do a better job at integrating and synthesizing this body of theory as we study microbial communities. 

Line 508- As worded, it suggests isolates were derived from environments rather than mosquitoes from diverse environments. Consider clarifying.

Line 509-516. Lack of systematic sampling is evident here. 

Line 551-554- This should be a feature of the main text. As a resource contribution, the authors should spend more time on the current accessibility of this resource and the vision to expand access to the scientific community.

---

## [Decision Letter · Decision Letter 2]

1 Sep 2024

Dear Dr Brettell,

Thank you for your patience while we considered your revised manuscript "Establishment and comparative genomics of a collection of mosquito-associated bacterial isolates - MosAIC (Mosquito-Associated Isolate Collection)" for consideration as a Methods and Resources at PLOS Biology. Your revised study has now been evaluated by the PLOS Biology editors, the Academic Editor, one of the original reviewers and an additional reviewer. 

Based on the reviews, the Academic Editor's assessment of your revision, and feedback from an additional reviewer regarding Reviewer #3’s previous concerns, there are still key points that must be addressed before we can proceed with acceptance. One major issue is the overestimation of the observational analysis of the existing dataset, which was not adequately revised or discussed, as requested by Reviewer #3. As also pointed out from the additional comment, which you can find below, the biological conclusions are not solid, and therefore must be deemphasized and properly discussed in the main text. Furthermore, the Academic Editor has highlighted Reviewer #1’s concern about the potential common origin of isolates from lab colonies, as mosquito colonies are often shared between labs. While we do not expect a full resolution of this issue, it is essential to acknowledge it in the discussion section; although it is not expected to fully address this question, we required that you mention this in the discussion. I would like to emphasize that without these adjusments in the text, we will not be able to move towards publication.

Finally, please also address the following editorial concerns:

a) We routinely suggest changes to titles to ensure maximum accessibility for a broad, non-specialist readership, and to ensure they reflect the contents of the paper. In this case, we would suggest a minor edit to the title, as follows. Please ensure you change both the manuscript file and the online submission system, as they need to match for final acceptance:

"MosAIC: an annotated collection of mosquito-associated bacteria with high-quality genome assemblies"

b) Please mention in the abstract that the bacterial isolates are publicly available as a resource to the community

c) Thank you for providing the scripts to generate the figures. However, please state clearly how to use the scripts and to which figure they are useful. According to our PLOS Data Policy, we require that all data be made available without restriction: http://journals.plos.org/plosbiology/s/data-availability. For more information, please also see this editorial: http://dx.doi.org/10.1371/journal.pbio.1001797

Please make sure that you supply the necessary data and scripts for the following figures:

Figure 2BCDE, 3, S1, S2, S4ABCD, S5AB, S6, S10, S14, S15

As well, as the tree files, or the necessary data to build them, for figures 2A, S7, S8, S9. S11, S12, S13

d) Please cite the location of the data clearly in all relevant main and supplementary Figure legends, e.g. “The data underlying this Figure can be found in S1 Data” or “The data underlying this Figure can be found in https://doi.org/10.5281/zenodo.XXXXX”

e) Many thanks for providing the underlying scripts in GitHub. However, because Github depositions can be readily changed or deleted, please make a permanent DOI’d copy (e.g. in Zenodo) and provide this URL in the manuscript and Data Availability Statement. I have asked my colleagues to include this request alongside their own.

f) Please ensure that your Data Statement in the submission system accurately describes where your data can be found and is in final format, as it will be published as written there.

g) Per journal policy, if you have generated any custom code during the course of this investigation, please make it available without restrictions upon publication. Please ensure that the code is sufficiently well documented and reusable, and that your Data Statement in the Editorial Manager submission system accurately describes where your code can be found.

Please note that we cannot accept sole deposition of code in GitHub, as this could be changed after publication. However, you can archive this version of your publicly available GitHub code to Zenodo. Once you do this, it will generate a DOI number, which you will need to provide in the Data Accessibility Statement (you are welcome to also provide the GitHub access information). See the process for doing this here: https://docs.github.com/en/repositories/archiving-

**IMPORTANT - SUBMITTING YOUR REVISION**

*Resubmission Checklist*

*Published Peer Review*

*PLOS Data Policy*

*Blot and Gel Data Policy*

Sincerely,

Melissa

Melissa Vazquez Hernandez, Ph.D.

Associate Editor

PLOS Biology

REVIEWS:

Reviewer #1: The authors have made sufficient revisions on this manuscript. It is well written and provides a major resource that is available to the research community. This data will help researchers create hypotheses about genes/gene clusters found in mosquito-associated bacteria. It will also provide a location for future researchers to submit isolates from diverse locations and mosquito species.

ADDITIONAL REVIEWER COMMENTS:

My main concern is that most of the paper is the description and comparative analysis of the genomes on the collection. Given the biased sampling and lack of an a priori design to perform these comparisons the conclusion cannot be solid. The authors alert for this bias and do not perform proper statistical analysis of these comparisons but, nonetheless extensively duel on it [figure 5 is very difficult to understand in terms of gene classification categories. I cannot really understand the figure]. These analyses would not grant publication by themselves and have very weak biological significance as they are. Mostly, they are based on lab isolates that are extremely similar. Even if they come from different labs they may have been shared recently between them. Core genome and shared genes analysis of these extremely similar bacteria is not very interesting. The third reviewer exposes this in slightly different words: “…rather than discussing weakly-substantiated scientific observations that at this stage remain relatively dubious, particularly those that generalize features of mosquito symbiont genomes and function.“ The authors addressed only later, specific comments of reviewer 3 on scientific meaning, and with basically cosmetic changes.

---

## [Editor Report · Decision Letter 3]

11 Oct 2024

Dear Dr Brettell,

Thank you for the submission of your revised Methods and Resources "MosAIC: an annotated collection of mosquito-associated bacteria with high-quality genome assemblies" for publication in PLOS Biology. On behalf of my colleagues and the Academic Editor, Mathilde Gendrin, I am pleased to say that we can in principle accept your manuscript for publication, provided you address any remaining formatting and reporting issues. These will be detailed in an email you should receive within 2-3 business days from our colleagues in the journal operations team; no action is required from you until then. Please note that we will not be able to formally accept your manuscript and schedule it for publication until you have completed any requested changes.

IMPORTANT: Many thanks for providing additional information of how to recreate the figures. However, we are still missing that in the figure legends you indicate where can the readers find the data. We require that the legends say at the end something like "All code and data to recreate figures can be found at...". I have asked my colleagues to include this request alongside their own.

PRESS

Sincerely, 

Melissa

Melissa Vazquez Hernandez, Ph.D., Ph.D.

Associate Editor

PLOS Biology
